# Uric Acid and Cortisol Levels in Plasma Correlate with Pre-Competition Anxiety in Novice Athletes of Combat Sports

**DOI:** 10.3390/brainsci12060712

**Published:** 2022-05-31

**Authors:** Luis Fernando Garcia de Oliveira, Tácito Pessoa Souza-Junior, Juliane Jellmayer Fechio, José Alberto Fernandes Gomes-Santos, Ricardo Camões Sampaio, Cristina Vasconcelos Vardaris, Rafael Herling Lambertucci, Marcelo Paes de Barros

**Affiliations:** 1Institute of Physical Activity and Sports Science (ICAFE), Cruzeiro do Sul University, Rua Galvão Bueno 868, São Paulo 01506-000, SP, Brazil; luis.oliveira@ceunsp.edu.br (L.F.G.d.O.); joseafgs@gmail.com (J.A.F.G.-S.); ricardocamoes@bol.com.br (R.C.S.); crisvardaris@gmail.com (C.V.V.); 2Physical Education Program, University Center of Nossa Senhora do Patrocínio (CEUNSP), Program in Physical Education, Rua Madre Maria Basília 965, Itu 13300-903, SP, Brazil; 3Research Group on Metabolism, Nutrition and Strength Training (GPMENUTF), Campus Politécnico, Federal University of Paraná (UFPR), Rua Coronel Francisco Heráclito dos Santos 210, Curitiba 81531-980, PR, Brazil; tacitojr@ufpr.br; 4Santos Football Club, Rua Princesa Isabel s/n, Santos 11075-500, SP, Brazil; jfechio@hotmail.com; 5Department of Human Movement Sciences, Federal University of São Paulo (UNIFESP), Rua Silva Jardim 136, Santos 11015-020, SP, Brazil; rlambertucci@unifesp.br

**Keywords:** oxidative stress, antioxidant, free radical, steroids, glucocorticoid, exercise, jiu-jitsu, depression, mood, motivation

## Abstract

Pre-competition anxiety is very prevalent in novice athletes, causing stress and drastic decreases in their performances. Cortisol plays a central role in the psychosomatic responses to stress and also in the physiology of strenuous exercise. Growing evidence links uric acid, an endogenous antioxidant, with oxidative stress and anxiety, as observed in many depressive-related disorders. We here compared anxiety inventory scores (BAI and CSAI-2), cortisol and biomarkers of oxidative stress in the plasma of novice combat athletes (white and blue belts) before and after their first official national competition, when levels of stress are presumably high. Although the novice fighters did not reveal high indexes of anxiety on questionnaires, significant correlations were confirmed between cortisol and cognitive anxiety (Pearson’s *r* = 0.766, *p*-value = 0.002, and a ‘strong’ Bayesian inference; BF10 = 22.17) and between pre-post changes of plasmatic uric acid and somatic anxiety (*r* = 0.804, *p* < 0.001, and ‘very strong’ inference; BF10 = 46.52). To our knowledge, this is the first study to report such strong correlations between uric acid and pre-competition anxiety in novice combat athletes. The cause-consequence association between these indexes cannot be directly inferred here, although the interplay between uric acid and anxiety deserves further investigation.

## 1. Introduction

Many psychological tests have been long applied to express the levels of anxiety in athletes before competition, but there is still scarce information linking the perception of anxiety and the physiological/metabolic manifestations under stressful conditions in sports [1]. The pre-competition anxiety usually results from a multitude of factors, such as exhaustive practice before tournaments (overtraining), psychological stress from oppressive coaches and/or family, personal low confidence, a perfectionistic personality, and hostile competition environments. Many researchers have been comparing hormonal responses, cortisol in particular, in training sessions or laboratory-based exercises versus real competitive circumstances, in order to understand the magnitude of anxiety and stress on athletic performance [2,3]. Interestingly, recent findings have shown that blunted cardiovascular and cortisol responses to acute psychological stress are actually associated with adverse behavioral and detrimental health outcomes. Depression, obesity, anorexic/bulimic patterns, and drug addictions have been long associated with the suboptimal functioning of the frontal-limbic systems in the human brain [4]. That particular brain region is associated with the regulation of the motivation behavior in the face of challenge (catecholamine-driven), as well as the decline of decision-cognitive capacity under adverse circumstances [5]. Accordingly, previous studies on stress responses in sports have suggested the importance of balance between psychological (e.g., rate of perceived exertion and anxiety) and physiological demands (e.g., heart rate and cortisol responses) for optimized performance [6,7]. Indeed, independently of the combat sports types (striking and/or grabbing), a recent systematic review showed that both official and simulated bouts add real stress to the hormonal system of practitioners, particularly expressed as the elevation of cortisol and (nor)epinephrine levels via the HPA axis [3]. In agreement, salivary cortisol levels in karate athletes were almost two-fold higher after two combat matches in competition than pre-competition [8]. Interestingly, wrestlers that actually lost their matches had higher levels of anticipatory cortisol and anxiety compared with winners in the same competition [9]. Updated reviews have been consistently examining the close relationship between stress hormones (mainly HPA-axis cortisol and epinephrine), anticipatory stress/anxiety, and a decrease in athletic performance [10].

Sports anxiety patterns can be dismantled into three main components: cognitive anxiety, somatic anxiety, and self-confidence. Cognitive anxiety is defined as the mental component of anxiety and is caused by negative expectations about success or by negative self-evaluation. Somatic anxiety refers to the physiological and affective elements of the anxiety experience that develop directly from autonomic arousal. Finally, self-confidence encompasses the general perception of achievement from the competitors [11]. Cortisol is a glucocorticoid steroid hormone produced by the adrenal cortex that plays a central role in the physiological and behavioral response to stress through the activation of the hypothalamic-pituitary-adrenocortical axis [12]. Cortisol stimulates protein catabolism in almost all animals (especially in skeletal muscles) for an additional provision of glucogenic amino acids for hepatic gluconeogenesis during exercise and starvation conditions. Moreover, cortisol triggers fat degradation by lipases in adipocytes, depresses inflammatory responses (which could be harmful regarding the risks of infections), affects the circulating number of lymphocytes and migrating macrophages, and diminishes the secretion of cytokines, causing lower post-exercise immune responsiveness [13]. Therefore, cortisol also modulates the extension of oxidative stress associated with exercise, which is a physiological condition imposed by imbalances between reactive oxygen/nitrogen species (ROS/RNS) production and antioxidant capacity in different cells and tissues [14]. A direct link between anxiety and oxidative stress still needs investigation, either under pathological or normal physiological conditions. The supplementation with different antioxidant compounds (carotenoids, omega-3 fatty acids, ascorbic acid, etc.) was consistently proven to induce anxiolytic effects in experimental animals [15,16].

Strong evidence demonstrates that uric acid, an endogenous antioxidant, is probably the major link between oxidative stress and anxiety, as observed in many depressive-related disorders [17]. Recent findings using functional magnetic resonance imaging (fMRI) suggest that uric acid levels (in saliva) are also related with hippocampal activity and behavioral stress, with minor indications in the prefrontal cortex or amygdala [18]. Uric acid is also an end-point compound of purine catabolism which, again, connects mood and behavior (especially during psychosocial stress) with redox/energy metabolism, since purine degradation is a metabolic pathway promptly activated in energy-depleted cells [19].

Therefore, the aim of this study was to compare anxiety indexes, cortisol levels and oxidative stress biomarkers in combat athletes under harsh, intimidating, and supposedly stressful conditions. For that purpose, we selected novice jiu-jitsu athletes, ranked as white and blue belts, before the first official competitions of their careers. We hypothesize that, due to their inexperience upon the adverse/hostile conditions imposed by their first official competitions (especially considering combat sports), novice athletes will show high indexes of anxiety which will match with cortisol levels and changes of oxidative stress biomarkers, especially uric acid, based on evidence from other psychiatric studies.

## 2. Materials and Methods

### 2.1. Subjects

Thirteen young, (22 ± 4 years-old), ranging from 18 to 26 years-old, male, college non-professional practitioners of jiu-jitsu participated in the study (N = 13). Jiu-jitsu is a popular martial art of which the main goal is the submission of the opponent through the application of a stranglehold or joint locks, among several other grabbing techniques, to gain points and win the match [20]. Regarding physiology demands, jiu-jitsu is characterized by high and low intensity-intermittent efforts involving isometric, concentric, and eccentric muscle contractions [21,22]. The competition mind setting is clearly characterized by aggressiveness, self-confidence, and intimidation [23]. All subjects had short experience with jiu-jitsu (<three years) but routinely trained specific techniques at night, four times/week, comprising a total of eight hours per week. All subjects were white and blue belt ranked athletes. Subjects were not taking any antioxidant or multivitamin supplements in the last two months and their diets did not contain more than 4000 cal/day or a protein intake > 1.75 g/kg body mass (Exclusion criterion 1). Exaggerated high protein/fat intake could affect hormonal levels and iron homeostasis which would interfere in oxidative stress indexes [24]. The subjects attested to no use of anti-inflammatory drugs in the last two months (Exclusion criterion 2). This study was carried out during an annual national jiu-jitsu competition, usually in mid-December, gathering around 200 competitors, from white to black belts (all categories involved). All the subjects were informed of the risks and benefits of the study prior to any data collection and gave written informed consent for their participation. The protocol of the study was approved by the Ethics Committee of the hosting institution (CE/UCS-019/2013) in accordance with the Declaration of Helsinki.

### 2.2. Psychological Tools

Individual appointments between athletes and the psychologist were held in a reserved area of the hotel lounge by 6:00 a.m., approximately 3 h before the beginning of competitions. Although we expected that the anxiety level would increase later on, logistics issues (meeting for coaching instructions, warming up, etc.) limited our further access to the athletes. We are aware of adapted stress/questionnaires to immediately evaluate pre-competition stress (e.g., CSAI-30), but we were not allowed to approach the athletes under these circumstances [25].

The Competitive State Anxiety Inventory-2 (CSAI-2) is a simple and valid instrument for the assessment of anxiety before an athletic competition [26,27]. The CSAI-2 was used here to measure pre-competition cognitive anxiety, somatic anxiety, and self-confidence among the novice athletes. The CSAI-2 comprises 27 items, with nine items in each subscale. Participants were asked to rate the intensity of each symptom on a scale from one (not at all) to four (very much so). Despite the fact that some CSAI-2 data analysis for Brazilian participants showed that the three-factor model was apparently inadequate, with marginal fit coefficients, the internal psychometric criteria here accomplished the requirements for all three factors: cognitive anxiety, somatic anxiety, and self-confidence [28]. CSAI-2 has been concurrently used with the evaluation of other physiological parameters (e.g., heart rate or cortisol) for a better characterization of psychophysiological responses to competitions in tennis and in other sports [7,29].

The Beck Anxiety Inventory (BAI) is a 21-item self-report inventory that measures the severity of anxiety in psychologic disturbed populations. Subjects rate the items according to how much they have been bothered by a particular symptom over the past week; each item is rated on a 4-point scale ranging from 0 (not at all) to 3 (severely—“I could barely stand it”). The total score range is between 0–63. BAI focuses predominantly on the physiological aspect of anxiety. The BAI test demonstrated good factorial validity, with somatic anxiety and subjective anxiety factors emerging as the internal psychometric properties evaluated here [30]. Four of the 21 items are anxious mood terms; three items assess specific fears; the remaining 14 items assess the symptoms of autonomic hyperactivity and motor tensions, generalized anxiety, disorder, and panic [31]. The BAI questionnaire was applied immediately after the CSAI-2, individually, in the same reserved place, also approximately 3 h before the beginning of competitions.

### 2.3. Blood Sampling

Blood samples were withdrawn from the forearm cubital vein of the subjects early in the morning of competition, at 7:00 a.m., approximately 2 h before their first combat. Samples were collected in EDTA-containing Vacutainer^®^ (Franklin Lakes, NJ, USA) tubes, and immediately centrifuged for 5 min at 4000 rpm (550× *g*, room temperature) to isolate serum and plasma. The same procedure was used to collect blood samples after combat, within 15 min after the completion of the fight. All samples were stored first in dry ice, and then at −80 °C until chemical analysis.

### 2.4. Biochemical Assays

All spectrophotometry determinations were performed in a 96-well microplate reader, a SpectraMax M5 (Molecular Devices, Sunnyvale, CA, USA). Commercial kits obtained from Abcam (Cambridge, MA, USA) were used to quantify glucose (#ab65333), total iron (#ab83366), and uric acid (#ab65344) concentrations in plasma. Serum cortisol was determined by the #500360 ELISA kit purchased by Cayman Chemicals (Ann Arbor, MI, USA). The antioxidant capacity of plasma was measured as its ferric-reducing activity (FRAP) [32]. The FRAP method quantifies metal ligands in plasma that form redox inactive complexes [Fe(L)]^n+^ that, thereby, limit Fenton-type reactions and the formation of prooxidants. The ferrous-chelating agent 2,4,6-tripyridyl-S-triazine (TPTZ) was here replaced by its analog 2,3-bis(2-pyridyl)-pyrazine (DPP) [33]. Briefly, 10–20 µL of samples were mixed with the FRAP reactant solution containing 10 mM DPP (from a stock solution in 40 mM HCl) and 20 mM of FeCl_3_ in a 0.30 M acetate buffer (pH 3.6). Absorbance at 593 nm was recorded for 4 min to determine the rate of Fe^2+^-DPP complex formation as compared to a standard curve.

Reduced and oxidized glutathione contents in plasma (GSH and GSSG, respectively) were measured by the reaction of reduced thiol groups (-SH) with 5,5′-dithiobis-2-nitrobenzoic acid (DTNB) to form 5-thio-2-nitrobenzoic acid (TNB), which is stoichiometrically detected by the absorbance at 412 nm [34]. GSH and GSSG standards were purchased from Sigma-Aldrich (Burlington, MA, USA). The reducing power of samples was calculated based on the ratio between reduced (GSH) and total glutathione (GSH + GSSG) in the plasma of subjects.

Lipid peroxidation was assayed as thiobarbituric acid-reactive substances (TBARS) in plasma [35]. The concentration of TBARS in plasma was measured after sample treatment with 4% butylated hydroxytoluene (BHT, in ethanol) and further reaction with 0.375% thiobarbituric acid in 0.25 M HCl and 1% Triton X-100 (15 min, at 100 °C). Malondialdehyde equivalents (nmol MDA/mg protein) were calculated based on absorbance at 535 nm against blanks lacking TBA, and using 1,1,2,2-tetroxyethylpropane as standard. Pre/post variations of all biochemical indexes (Δx) were also calculated by dividing the individual values obtained after competition (x_post_) by those obtained before (x_pre_), for each index and subject.

### 2.5. Statistical Analysis

Biochemical determinations were presented as (mean ± SE) and were analyzed by the paired t-test, used for normalized population distributions (determined by Kolmogov-Smirnov test), or by Wilcoxon test for a nonparametric statistical index. Significance was assumed when *p* < 0.05. Linear correlations between biochemical determinations were tested by calculating Spearman’s rho factor for *p* < 0.05. Effect size (Cohen’s d factor) and confidence intervals were calculated with OriginPro 2016 Sr2 software (Northampton, MA, USA) and the post-hoc classification for effect size (based on modular Cohen’s d values) was: |d| ≥ 0.10, very small; |d| ≥ 0.20, small; |d| ≥ 0.50, medium; |d| ≥ 0.80, large; |d| ≥ 1.20, very large; and |d| ≥ 2.00, huge [36]. Bayesian inference analysis was performed based on prior information collected from published works, considering: BF10 < 3 (anecdotal); 10 > BF10 > 3 (moderate); 30 > BF10 > 10 (strong); 100 > BF10 > 30 (very strong); 300 > BF10 > 100 (extremely strong); BF10 > 300 (extreme) [37,38].

## 3. Results

### 3.1. Absolute Scores

CSAI and BAI tests were applied early in the morning of the first day of their official competition. After the psychological interviews, blood samples were collected for biochemical analysis. Table 1 shows the CSAI scores for somatic anxiety, cognitive anxiety, and self-confidence, and the background levels of cortisol (µg/dL) in serum of novice athletes. Table 2 presents BAI scores, indicating an overall minimum to slight degree of anxiety from the group of novice athletes.

Regarding biochemical indexes, no clear evidence of oxidative stress was observed in the plasma of novice jiu-jitsu athletes, as observed by pre/post variations related to their first competition. Table 3 shows no significant changes of the overall antioxidant capacity (FRAP assay), redox balance (reducing power), lipid oxidation (TBARS levels), and uric acid concentrations in plasma, although most Bayesian inferences, based on previous works, were from anecdotal (BF < 3) to moderate (10 > BF > 3). All parameters in Table 3 showed small to medium effect sizes (0.2 < |d| < 0.5). On the other hand, some parameters have presented significant differences in plasma of novice jiu-jitsu athletes pre/post their first competition (Figure 1), such as glucose (*p*-value < 0.001, very large to huge effect size Cohen’s |d| = 1.6590, and extremely strong Bayesian inference BF10 = 852.17), Fe (*p*-value = 0.003, very large effect size Cohen’s |d| = 1.0433, and very strong Bayesian inference BF10 = 34.014), GSH (*p*-value = 0.042, small to medium effect size Cohen’s |d| = 0.630, and moderate Bayesian inference BF10 = 3.557), and GSSG (*p*-value = 0.048, small effect size Cohen’s |d| = 0.4412, and moderate Bayesian inference BF10 = 7.939).

### 3.2. Correlations

As shown in Table 4 and Figure 2, linear correlations and Bayesian inferences were calculated between indexes of competition anxiety (from CSAI and BAI tests), the individual levels of cortisol and oxidative stress biomarkers before competition. Cortisol levels were linearly correlated with cognitive anxiety, as estimated by CSAI tests: Pearson’s *r* = 0.766, *p*-value = 0.002, and a strong Bayesian inference; BF10 = 22.17. Regarding the inventories, CSAI-somatic anxiety was both correlated with CSAI-cognitive anxiety (Pearson’s *r* = 0.762, *p*-value = 0.002, and a strong Bayesian inference; BF10 = 20.68) and with BAI scores (Pearson’s *r* = 0.655, *p*-value = 0. 015, and a moderate Bayesian inference; BF10 = 4.877). On the other hand, no correlations were observed between anxiety scores from CSAI or BAI tests or even with cortisol levels in plasma with oxidative stress biomarkers before competition, indicating athletes’ basal condition on that morning preceding their participation. Expected correlations were observed between oxidative stress biomarkers in plasma measured before competition: TBARS versus GSH (*r* = 0.590, *p*-value = 0.034, although with anecdotal Bayesian inference; BF10 < 3), TBARS versus GSSG (*r* = 0.756, *p*-value = 0.003, with strong Bayesian inference; BF10 = 19.00), FRAP versus GSSG (*r* = 0.564, *p*-value = 0.045, with anecdotal Bayesian inference; BF10 < 3), glucose versus reducing power (*r* = 0.736, *p*-value = 0.004, with strong Bayesian inference; BF10 = 13.75), glucose versus iron content (*r* = 0.696, *p*-value = 0.008, with moderate Bayesian inference; BF10 = 7.930), and uric acid versus iron content (*r* = −0.599, *p*-value = 0.031, with anecdotal Bayesian inference; BF10 < 3).

The pre/post variations of oxidative stress biomarkers (Δx indexes in Table 5) were correlated between them and with the anxiety scores from CSAI and BAI tests. From all tests performed, the pre/post variations of uric acid content in plasma of individuals was surprisingly well correlated, with almost all psychological parameters evaluated, except CSAI-self-confidence (Table 5): ΔUric acid versus CSAI-somatic anxiety (Pearson’s *r* = 0.804, *p*-value < 0.001, and a strong Bayesian inference; BF10 = 46.52), ΔUric acid versus CSAI-cognitive anxiety (Pearson’s *r* = 0.648, *p*-value = 0.017, and a moderate Bayesian inference; BF10 = 4.526), ΔUric acid versus BAI (Pearson’s *r* = 0.585, *p*-value = 0.036, although an anecdotal Bayesian inference; BF10 < 3). Among oxidative stress biomarkers, we only observed significant correlations between ΔGlucose versus ΔReducing power (Pearson’s *r* = 0.563, *p*-value = 0.045, and an anecdotal Bayesian inference; BF10 < 3).

## 4. Discussion

In contrast to our original hypothesis, the novice jiu-jitsu athletes did not declare high levels of anxiety anticipating their official competition, based on CSAI-2 and BAI scores. On the other hand, significant correlations were confirmed between cognitive anxiety scores (CSAI-2) and plasma cortisol and, especially, between cognitive and somatic anxiety and pre/post changes of uric acid (Δuric acid) related to their participation in the competition. Many authors have reported that cortisol, as a suitable indicator of emotional stress, responds directly to the competition environment, and according to friendly (home) or hostile (opponent field), during different sport events [39,40]. Souza et al. (2018) also observed high levels of salivary cortisol responses and emotional stress in the pre-competition period, as reported by self-confidence scores [26]. Accordingly, a recent pilot study concluded that a mindfulness-based intervention was positively associated with a diminished physiological (measured by salivary cortisol and α-amylase) and psychological stress response (based on CSAI scores) to competition, which corroborates the cross-link between these factors [41]. As recently observed in a meta-analysis study, strong associations were observed between competitive anxiety and gender, lower age, and, especially, inexperience in that particular sport, and it was not necessarily limited to combat activities [42]. Our results showed that novice jiu-jitsu competitors attested minimum to slight levels of somatic and cognitive anxiety before competition, whereas self-confidence was evaluated as moderate to high (2.69 ± 0.61, scores from 0 to 4). Moreover, cognitive and somatic anxiety indexes extracted from the same CSAI inventory were highly correlated (*p* = 0.002 with strong Bayesian inference; Table 4), whereas only CSAI-cognitive anxiety showed correlation with BAI results (*p* = 0.015 and moderate Bayesian inference).

Self-confidence has been controversially associated with sports performance, and its validation as a realistic dimension of anxiety (extracted from CSAI inventory) is again under debate [43]. Worthy of note, the signaling from dopamine neurons of the ventral tegmental area (VTA) in the mesoaccumbal and mesocortical pathways of murine models has been shown to play a vital role in anxiety behavior [44]. The ventral tegmental area (VTA) is best known for its robust dopaminergic projections to forebrain regions and their critical role in regulating reward, motivation, cognition, and aversion [45]. Therefore, several studies are now focusing on dopaminergic circuits (also in VTA segments) to address the self-confidence component of sports anxiety, in parallel to those cortisol-mediated processes [46]. The validation of self-confidence as an accurate dimension of competitive anxiety has been particularly discussed in combat sports, since the statement of ‘confidence’ is usually present as a renowned prerequisite among combat athletes to suggest their readiness to begin the competition, despite the real insecurity (and fear) under those circumstances [47]. Perfectionism is apparently a relevant inherent element of the low self-confidence of athletes before competition. Perceptions of parental and coach pressure aligned with exacerbated concern over mistakes, again emerged as especially important issues on anxiety behavior [48]. Unfortunately, we did not evaluate the athletic performance of our novice jiu-jitsu athletes in the official competition for a further comparison with their attested CSAI scores. At the present time, many studies have already reported significant correlations of the CSAI and BAI scores with salivary and plasmatic levels of cortisol in athletes anticipating competition [49].

Cortisol and anxiety association was strongly dependent on the related sport, considering individual (e.g., track and field sports) or collective types (basketball, soccer, etc.), involving physical contact (e.g., rugby, judo) or not (volleyball, tennis), requiring different levels of aggressiveness (e.g., mixed martial arts, karate) or deep concentration (shooting and archery, for example) [42]. Although somatic anxiety hypothetically reflects the physiological (endocrine) manifestation of anxiety in subjects, we could only observe significant correlations between plasma cortisol levels and cognitive anxiety in our novice jiu-jitsu athletes before competition (Table 4). Self-confidence was not significantly correlated, either. Similar results were observed in judoists during regional and national competitions, as positive correlations were clearly observed between cognitive- and somatic-anxiety CSAI scores and plasma cortisol levels [7], as well as in canoeing, runners and jiu-jitsu athletes [26].

Although an increase of cortisol appears to be important while preparing for optimized mental and physical competitive control [50], extreme elevations in cortisol may lead to a severe decline in immune responsiveness after physical effort, e.g., during recovery periods, which affects competitiveness, mood, and the longevity of an athlete’s career, as a long-term effect [51]. As an endocrine regulator, the steroid hormone cortisol also affects protein and energy metabolism, which influences immune competence [52]. Therefore, it is expected that the stress hormone cortisol would also have influence on redox metabolism [14].

At rest, background levels of many biomarkers of oxidative stress in plasma were moderately to strongly correlated, congruent with a normalized redox balance in plasma before competition. Lipid oxidation levels (TBARS) were well correlated with both GSH and GSSG contents (Table 4), which reinforces the key participation of thiol-dependent systems on sustaining efficient antioxidant capacity in plasma to limit oxidative modifications on essential biomolecules [53]. As a major endogenous antioxidant in plasma, uric acid was also correlated with iron content in the plasma of subjects before competition (Table 4). Urate (deprotonated uric acid) forms a coordination complex with ferric and ferrous ions (lower affinity for Fe^2+^), preventing the metal participation as a redox-catalyst of ROS/RNS formation [54]. Similar correlations between uric acid and iron ions were previously reported by our group in oxidative stress conditions imposed to trained male young subjects by an exhausting Wingate test [37].

Reinforcing the key antioxidant role of thiol compounds in biological systems, variations of GSH and GSSG levels following the jiu-jitsu competition were apparently able to cope with the 85% increase of iron content (prooxidant) in plasma, which sustained an adequate redox balance in the plasma of subjects. Accordingly, reducing power indexes and TBARS levels were unaltered after competition (Table 3). Bayesian analyses supported by previous studies from our and other groups reveal moderate inferences (3 < BF < 10) of lipid oxidation (TBARS) and the antioxidant capacity of plasma (FRAP), despite the fact that the *p*-values did not indicate significance. Uric acid levels in plasma were also unaltered pre/post competition (Table 3). It is worthy to note that it has been previously reported that uric acid, in fact, shows a late onset accumulation in plasma (>1 h), which serves as a second efficient wave of antioxidant activity in the system, aiming to prolong the balanced redox conditions in plasma following strenuous exercises [37]. The main antioxidants in plasma during or immediately after exercises are GSH, lipid-soluble tocopherols (mostly associated with circulating lipoproteins) and ascorbic acid [55].

The pre/post variations of oxidative stress biomarkers here, expressed as Δx (x for each biomarker), are supposedly more advantageous for the monitoring of the metabolic adaptations upon the oxidative challenge imposed by exercise and physical activity. The pre/post analysis should, thus, minimize the individual component in each of those indexes which prevail when absolute values are evaluated instead. From all calculations performed, ΔUric acid showed the highest correlation indexes with most anxiety scores extracted from CSAI and BAI inventories (Table 5). The correlation between ΔUric acid and CSAI-cognitive anxiety showed a Pearson’s factor *r* = 0.804, *p* < 0.001, and with very strong Bayesian inference. Many authors have reported that uric acid peaks were shown to depend on the exercise intensity and to the individual muscle fiber composition of competing subjects, which could, therefore, be affected by fitness, body adaptations to exercise, and previous physiological conditions [56].

Uric acid is the endpoint product of purine catabolism, which is a pathway activated in muscle fibers by higher AMP/ATP ratios (energy depletion), and in endothelial cells upon increased blood circulation [57]. As mentioned earlier, uric acid apparently counteracts the prooxidant action of Fe^2+/3+^ ions released in plasma by the exercise-induced rupture of iron homeostasis, and was shown to correlate with plasma creatine kinase activity, a renowned biomarker of muscle injury [58]. Interestingly, associations between uric acid levels in plasma and affective disorders, such as depression and anxiety, were observed in patients with current (N = 1648), or remitted (N = 609) depression and/or anxiety disorders, compared to control (N = 618). In general, depressive, anxiety and phobic symptom severity, as well as symptom duration, were negatively associated with uric acid concentration in plasma [59]. Recent meta-analysis studies confirmed the lower uric acid levels in plasma of depressive/anxiety patients [60], which were reverted upon antidepressant treatment [61].

Although we did not observe a direct correlation between background levels of uric acid in plasma (pre) and anxiety-depressive behavior in our subjects, the interdependence between those factors was significantly attested when the pre-post variation of uric acid in plasma was accessed. When oxidative stress indexes were approached as a pre/post variation, we actually minimized the individual influence on those variables, and the correlation became solid. It is still unclear if the physiological condition of subjects (fitness and conditioning), or experience in sports would affect the correlation between cognitive/somatic anxiety and uric acid changes in plasma, or if other indexes would also show relevant interdependence.

## 5. Conclusions

To the best of our knowledge, this is the first study to report strong correlations between uric acid levels in novice combat athletes in the beginning of their careers (first official competitions) and anxiety levels, which were supposedly high under those circumstances. Therefore, plasmatic uric acid might be applied as a potential biomarker of the anticipatory stress (approaching all anxiety dimensions) experienced by combat athletes before competitions, in parallel with psychological questionnaires. Based on that, sports psychologists, coaches, and sport physicians would be able to adapt training/polishing protocols, mindfulness sessions and many other focusing procedures to minimize stress, particularly on their more susceptible athletes before competitions. It should be kept in mind that we could not demonstrate a significant correlation between cortisol and uric acid (or their pre/post variations). This interventional study was conducted with a limited number of participants and the cause-effect association between uric acid and psychopathology of anxiety in combat sports cannot be directly inferred here, but this aspect is definitely worthy of further investigation.

## Figures and Tables

**Figure 1 brainsci-12-00712-f001:**
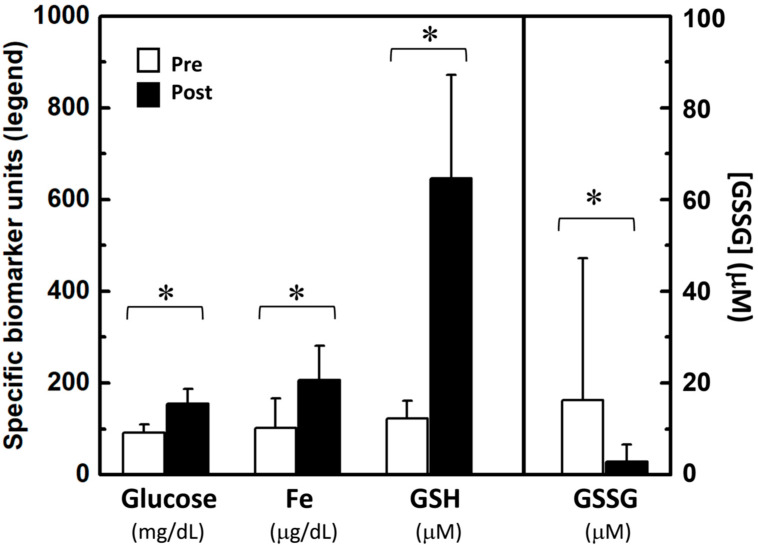
Concentrations of glucose, total iron (Fe), reduced and oxidized glutathione (GSH and GSSG, respectively) in plasma of novice jiu-jitsu athletes before (pre) and after (post) their first official competitions. (* *p* < 0.05; N = 13).

**Figure 2 brainsci-12-00712-f002:**
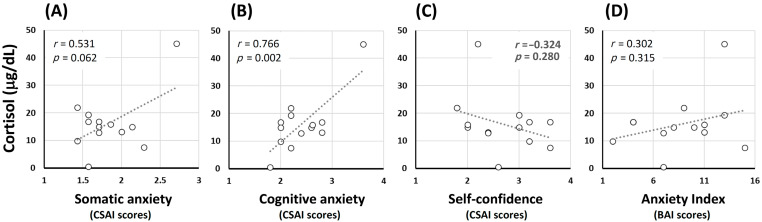
Linear correlation between cortisol levels and (**A**) somatic anxiety; (**B**) cognitive anxiety; or (**C**) self-confidence CSAI scores; and (**D**) anxiety BAI scores of novice jiu-jitsu athletes in their first official competitions (N = 13).

**Table 1 brainsci-12-00712-t001:** CSAI scores and cortisol determination in plasma of novice jiu-jitsu athletes before an official competition (N = 13).

Parameter	Somatic Anxiety	Cognitive Anxiety	Self-Confidence	Cortisol (µg/dL)
Pre-combat ^1^	1.82 ± 0.37	2.40 ± 0.48	2.69 ± 0.61	16.1 ± 10.3

^1^ data are presented as mean ± SD.

**Table 2 brainsci-12-00712-t002:** BAI scores of novice jiu-jitsu athletes before an official competition (N = 13).

	Minimum	Slight	Moderate	High	Maximum
Number subjects	08	05	0	0	0
BAI scores ^1^	6.4 ± 2.8	12.6 ± 1.7	0	0	0

^1^ data are presented as mean ± SD.

**Table 3 brainsci-12-00712-t003:** Pre/post biochemical parameters in plasma of novice jiu-jitsu athletes in their first official competitions (N = 13).

Parameter	Pre	Post	*p*-Value	Cohen’s d	BF
Uric acid ^1^	5.69 ± 0.53	5.78 ± 0.96	0.686	−0.1150	3.334
TBARS ^2^	3.77 ± 0.66	3.14 ± 1.33	0.187	0.3879	7.452
FRAP ^3^	2.06 ± 2.09	3.66 ± 5.49	0.355	−0.2668	6.275
Reducing power ^4^	0.757 ± 0.234	0.851 ± 0.212	0.179	−0.3954	1.574

^1^ expressed in mg/dL; ^2^ expressed in nmol/mL; ^3^ expressed in nmolFe^2+^/min/mL; ^4^ dimensionless.

**Table 4 brainsci-12-00712-t004:** Correlation matrix between anxiety scores, cortisol and biomarkers of oxidative stress in plasma of novice jiu-jitsu athletes before their first official competitions. (N = 13).

		Cortisol	Somatic (CSAI)	Cognitive (CSAI)	SC (CSAI)	BAI	GSH	GSSG	Reducing Power	TBARS	FRAP	Glucose	Fe	Uric Acid
Cortisol	Pearson *r*	----												
	*p*-value	----												
	BF10	----												
Somatic (CSAI)	Pearson *r*	0.531	----											
	*p*-value	0.062	----											
	BF10	1.659	----											
Cognitive (CSAI)	Pearson *r*	0.766 **	0.762 **	----										
	*p*-value	0.002	0.002	----										
	BF10	22.17 ^‡‡^	20.68 ^‡‡^	----										
SC (CSAI)	Pearson *r*	−0.324	−0.182	−0.419	----									
	*p*-value	0.280	0.552	0.154	----									
	BF10	0.581	0.401	0.858	----									
BAI	Pearson *r*	0.302	0.655 *	0.380	−0.281	----								
	*p*-value	0.315	0.015	0.200	0.352	----								
	BF10	0.540	4.877 ^‡^	0.719	0.506	----								
GSH	Pearson *r*	0.152	0.156	0.185	−0.153	0.095	----							
	*p*-value	0.621	0.610	0.544	0.619	0.758	----							
	BF10	0.381	0.384	0.403	0.382	0.356	----							
GSSG	Pearson *r*	−0.106	−0.350	−0.224	0.266	−0.242	0.324	----						
	*p*-value	0.732	0.241	0.463	0.380	0.425	0.281	----						
	BF10	0.360	0.637	0.436	0.485	0.456	0.579	----						
Red. Power	Pearson *r*	0.200	0.400	0.292	−0.391	0.244	0.426	−0.386	----					
	*p*-value	0.512	0.176	0.334	0.187	0.421	0.147	0.193	----					
	BF10	0.415	0.785	0.522	0.754	0.458	0.886	0.738	----					
TBARS	Pearson *r*	−0.108	<−0.001	0.022	0.043	0.126	0.590 *	0.756 **	0.008	----				
	*p*-value	0.727	0.999	0.943	0.888	0.683	0.034	0.003	0.979	----				
	BF10	0.361	0.341	0.342	0.344	0.368	2.609	19.00 ^‡‡^	0.341	----				
FRAP	Pearson *r*	−0.158	−0.432	−0.130	0.487	−0.499	−0.116	0.564 *	−0.424	0.255	----			
	*p*-value	0.605	0.141	0.671	0.092	0.082	0.705	0.045	0.149	0.400	----			
	BF10	0.385	0.915	0.370	1.240	1.342	0.364	2.113	0.861	0.471	----			
Glucose	Pearson *r*	0.008	0.277	−0.074	−0.355	0.226	0.183	−0.454	0.736 **	−0.257	−0.552	----		
	*p*-value	0.980	0.359	0.811	0.234	0.459	0.549	0.120	0.004	0.397	0.050	----		
	BF10	0.341	0.500	0.350	0.650	0.438	0.402	1.026	13.75 ^‡‡^	0.473	1.934	----		
Fe	Pearson *r*	0.045	0.227	−0.022	−0.106	−0.134	0.208	0.064	0.538	−0.021	−0.130	0.696 **	----	
	*p*-value	0.884	0.455	0.944	0.731	0.661	0.496	0.835	0.058	0.945	0.671	0.008	----	
	BF10	0.344	0.440	0.342	0.360	0.372	0.421	0.348	1.745	0.342	0.370	7.930 ^‡^	----	
Uric acid	Pearson *r*	−0.383	−0.061	−0.146	0.071	0.250	0.027	0.106	−0.194	0.469	−0.151	−0.388	−0.599 *	----
	*p*-value	0.197	0.843	0.635	0.817	0.410	0.932	0.731	0.526	0.106	0.622	0.191	0.031	----
	BF10	0.727	0.347	0.378	0.349	0.465	0.342	0.360	0.410	1.117	0.381	0.744	2.828	----

* *p* < 0.05, ** *p* < 0.01, ^‡^ 10 > BF10 > 3 (Moderate), ^‡‡^ 30 > BF10 > 10 (Strong).

**Table 5 brainsci-12-00712-t005:** Correlation matrix between anxiety scores from CSAI or BAI tests or cortisol levels and the pre/post variation (Δx) of biomarkers of oxidative stress in plasma of novice jiu-jitsu athletes during their first official competitions. (N = 13).

		Cortisol	Somatic (CSAI)	Cognitive (CSAI)	SC (CSAI)	BAI	ΔGSH	ΔGSSG	ΔRed. power	ΔTBARS	ΔFRAP	ΔGlucose	ΔFe	ΔUric Acid
ΔGSH	Pearson *r*	−0.247	0.360	0.105	−0.271	0.108	----							
	*p*-value	0.415	0.226	0.734	0.370	0.726	----							
	BF10	0.462	0.664	0.360	0.492	0.361	----							
ΔGSSG	Pearson *r*	−0.247	0.337	0.257	−0.247	0.209	0.252	----						
	*p*-value	0.415	0.261	0.397	0.417	0.493	0.406	----						
	BF10	0.462	0.606	0.473	0.461	0.423	0.467	----						
ΔRed. Power	Pearson *r*	−0.176	−0.133	−0.295	0.008	0.206	0.483	−0.142	----					
	*p*-value	0.566	0.664	0.329	0.979	0.499	0.095	0.643	----					
	BF10	0.397	0.372	0.527	0.341	0.420	1.210	0.376	----					
ΔTBARS	Pearson *r*	0.064	−0.287	−0.044	0.174	−0.207	−0.079	0.282	0.062	----				
	*p*-value	0.836	0.343	0.886	0.569	0.498	0.798	0.351	0.839	----				
	BF10	0.348	0.514	0.344	0.395	0.420	0.351	0.507	0.347	----				
ΔFRAP	Pearson *r*	0.451	0.234	0.140	0.101	−0.080	−0.192	0.245	0.256	−0.004	----			
	*p*-value	0.122	0.442	0.648	0.744	0.795	0.529	0.420	0.399	0.989	----			
	BF10	1.009	0.447	0.375	0.358	0.352	0.409	0.459	0.472	0.341	----			
ΔGlucose	Pearson *r*	0.032	0.082	−0.055	0.385	0.249	0.288	−0.133	0.563 *	−0.135	0.374	----		
	*p*-value	0.917	0.791	0.858	0.194	0.412	0.339	0.664	0.045	0.661	0.208	----		
	BF10	0.343	0.352	0.346	0.735	0.464	0.517	0.372	2.099	0.372	0.701	----		
ΔFe	Pearson *r*	−0.090	−0.189	0.088	−0.282	0.142	0.236	0.028	0.130	0.130	−0.191	−0.029	----	
	*p*-value	0.769	0.536	0.775	0.350	0.643	0.438	0.927	0.672	0.673	0.533	0.924	----	
	BF10	0.355	0.406	0.354	0.508	0.376	0.449	0.342	0.370	0.370	0.407	0.342	----	
ΔUric acid	Pearson *r*	0.360	0.804 **	0.648 *	−0.080	0.585 *	0.281	0.293	−0.154	−0.371	−0.017	0.218	−0.273	----
	*p*-value	0.226	<0.001	0.017	0.795	0.036	0.352	0.332	0.615	0.213	0.957	0.475	0.367	----
	BF10	0.664	46.52 ^‡‡^	4.526 ^‡^	0.352	2.506	0.506	0.524	0.383	0.692	0.341	0.430	0.494	----

* *p* < 0.05, ** *p* < 0.01, ^‡^ 10 > BF10 > 3 (Moderate), ^‡‡^ 30 > BF10 > 10 (Strong)

## Data Availability

Original data is available with the corresponding authors M.P.B. and T.P.S.J., but not archived in databases elsewhere.

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
