# Peer review of "Uric Acid and Cortisol Levels in Plasma Correlate with Pre-Competition Anxiety in Novice Athletes of Combat Sports"

_brainsci, 2022, doi:10.3390/brainsci12060712_

Round 1
Reviewer 1 Report
Dear All,
This article is very novel and I think that this paper can add several points to the psychophysiological response of competitive anxiety. Thank you for your manuscript. I feel that you need some minor revisions for better quality. please see below:
in the introduction, you should add a paradigm for literature and previous studies.
please add some relevant studies for competitive anxiety-cortisol, anxiety and uric acid or SO, and explain the necessity of psychophysiological assessments in competition.
in the method, you should add the internal psychometric properties of the CSAI-2 and BAI
Why do you use the CSAI-2 and BAI 3 h before the beginning of competitions?do you have any references? I know that we have a time window before the competition and we should use the CSAI 30 at least 30 min before the competition?
the result part is ok and accurate.
in conclusion, you must compare the result with the previous studies and then explain the mechanisms. Please change this part and add the Strengths and limitations of the study.
Thank you for your attention
Author Response
Dear All,
This article is very novel and I think that this paper can add several points to the psychophysiological response of competitive anxiety. Thank you for your manuscript. I feel that you need some minor revisions for better quality. please see below:
ANSWER: First of all, I would like to thank the reviewer for the positive appreciation and compliments on our manuscript.
In the introduction, you should add a paradigm for literature and previous studies.
please add some relevant studies for competitive anxiety-cortisol, anxiety and uric acid or SO, and explain the necessity of psychophysiological assessments in competition.
ANSWER: As recommended by reviewer #1, we included an additional paragraph in the Introduction for a better presentation of the conection between competitive anxiety and cortisol. The inserted text is presented below:
“(lines 56-66) Indeed, independently of the combat sports types (striking and/or grabbing), a recent systematic review showed that both official and simulated bouts input real stress in the hormonal system of practitioners, particularly expressed as elevation of cortisol and (nor)epinephrine levels via the HPA axis [8]. In agreement, salivary cortisol levels in karate athletes were almost two-fold higher after their second combat in competi-tion compared to that recorded at pre-combat 1 [9]. Interestingly, loser wrestling fighters had higher levels of anticipatory cortisol and anxiety scores compared with winners in the same competition [10]. Updated reviews have been consistently ap-proaching the close relationship between stress hormones (mainly HPA-axis cortisol and epinephrine), anticipatory stress/anxiety, and decrease in athletic performance [11]”.
In the method, you should add the internal psychometric properties of the CSAI-2 and BAI.
ANSWER: As recommended by reviewer #1, we properly included more detailed description of the CSAI-2 and BAI questionnaires in this updated version of our manuscript. The included sentences were:
(lines 145-148): “Despite some CSAI-2 data analysis for Brazilian participants showed that the three-factor model was apparently inadequate, with marginal fit coefficients, the in-ternal psychometric criteria here accomplished all the three factors: cognitive anxiety, somatic anxiety, and self-confidence [29].”
(lines 157-159): “The BAI test demonstrated good factorial validity, with somatic anxiety and subjective anxiety factors emerging as the internal psychometric properties evaluated here [31]”.
Why do you use the CSAI-2 and BAI 3 h before the beginning of competitions? Do you have any references? I know that we have a time window before the competition and we should use the CSAI 30 at least 30 min before the competition?
ANSWER: Due to logistic issues, we planned to run the pre-competition evaluations at the hotel lobby at 6AM, before blood sampling and breakfast. We fully agree that the CSAI-2 scores collected 30 minutes before their inaugural combats would better reproduce their real level of stress, but, on the other hand, could still include artifacts, since athletes started their competitions at different times. Therefore, we assumed that, by evaluating all athletes at the same time, we could perceive a general aspect of the psychophysiologic stress experienced by them at that day, as a whole group, even if the maximum stress would be experienced later on. We agree that this is a very relevant question to be discussed and we decided to include this comments in the updated version of our manuscript:
(lines 133-139): “We decided to apply the psychophysiological questionnaires much earlier than their inaugural combats due to logistic issues, and also to capture the general psychophysi-ological stress experienced by them at that day, as a whole group, even if the maxi-mum stress would be experienced later on. Indeed, adapted stress/questionnaires were formulated for immediately pre-competition stress (e.g. CSAI-30), but these tools could also present artifacts here, since combat athletes usually start their fights at different times of the same day of competition [26]”.
The result part is ok and accurate.
ANSWER: Thank you for your appreciation.
In conclusion, you must compare the result with the previous studies and then explain the mechanisms. Please change this part and add the Strengths and limitations of the study. Thank you for your attention
ANSWER: We clearly understand the concerns of reviewer #1 and following suggestions, we included additional information in the discussion section here:
(lines 300-306): “Souza et al. (2018) also observed high levels of salivary cortisol responses and emo-tional stress in the pre-competition period, specially reported by self-confidence scores [42]. Accordingly, a recent pilot study concluded that a mindfulness-based interven-tion was positively associated with a diminished physiological (measured by salivary cortisol and a-amylase) and psychological stress responses (based on CSAI scores) to competition, which corroborates the cross-link between these factors [43]”.

Reviewer 2 Report
Thank you for the opportunity to revise the manuscript entitled "Uric Acid and Cortisol Levels in Plasma Correlate with Pre- 2 Competition Anxiety in Novice Athletes of Combat Sports"
The study investigated pre-competitive anxiety through biochemical and psychosocial methods. Specifically, the authors analyse the possible differences between anxiety's physiological and psychological aspects before an important match.
Here there are my comments about the research:
1) One of the main concerns is the entire submission. The article was submitted to Brain Science, and I expected an introduction and conclusions with a neuroscientific focus.
I think an almost complete rewrite of the introduction is necessary. Also, the discussion section should be rewritten to direct the focus on neuroscience (https://www.mdpi.com/journal/brainsci/about).
Discussion section is very articulated; however, as previously said, I did not find a neuroscientific focus, and in some parts, it seems to be a repetition of the results section. Moreover, it is important to emphasise what your research adds to the scientific community or experts in combat sports. In other words, how are these results significant for psychologists or coaches?
It seems you reported the limitations in the text; however, I suggest creating a new section to discuss the possible limitations of the study.
Minor points
The procedures are integrated into Psychological Analysis and Blood sampling section. I wondered if it is better to create a new section where you can explain the procedures.
For instance, could the "Biochemical assays" section be set as a subchapter of the Blood sampling section?
I would change Psychological Analysis to Psychological Tools (instruments).
Author Response
Comments and Suggestions for Authors
Thank you for the opportunity to revise the manuscript entitled "Uric Acid and Cortisol Levels in Plasma Correlate with Pre- 2 Competition Anxiety in Novice Athletes of Combat Sports"
The study investigated pre-competitive anxiety through biochemical and psychosocial methods. Specifically, the authors analyse the possible differences between anxiety's physiological and psychological aspects before an important match.
Here there are my comments about the research:
1) One of the main concerns is the entire submission. The article was submitted to Brain Science, and I expected an introduction and conclusions with a neuroscientific focus.
I think an almost complete rewrite of the introduction is necessary. Also, the discussion section should be rewritten to direct the focus on neuroscience (https://www.mdpi.com/journal/brainsci/about). Discussion section is very articulated; however, as previously said, I did not find a neuroscientific focus, and in some parts, it seems to be a repetition of the results section.
ANSWER: First of all, we would like to thank reviewer #2 for his/her evaluation of our work. We clearly understand reviewer #2 concerns about matching the scientific approach of our study with the journal’s scope. Undoubtedly, our contribution here is in the boundary of psychological studies and biochemistry/physiology of exercise. Based on the description of Brain Sciences aims: “Our aim is to encourage scientists to publish their experimental and theoretical results in as much detail as is required to fully convey the information”, specially because “Molecular and cellular neuroscience” is among the major topics of journal’s scope. Therefore, we were definitely encouraged to present here our molecular/biochemical perspective of the pre-competition anxiety in novice combat sport athletes. We obviously understand that we cannot simply present the redox mechanisms, cellular signaling pathways, etc. for Brain Sciences readers here, but rather, adapt our presentation for a more broad audience. Nevertheless, in respect to the comments and suggestions of reviewer #2, we revised both Introduction and Conclusion sections and attempted to improve the quality and the neuroscentific approach of our study.
(lines 47-53): “Interestingly, recent findings have shown that blunted cardiovascular and cortisol re-sponses to acute psychological stress are actually associated with adverse behavioral and health outcomes, such as depression, obesity, anorexic/bulimic patterns, and drug addictions, as a result of suboptimal functioning of fronto-limbic systems in human brain [4]. That particular brain region is associated with regulation of the motivated behavior in the face of challenge (catecholamine-driven) and the decline of decision-cognitive capacity [5]”.
(lines 325-334): “Self-confidence has been controversially associated with sports performance and its validation as a realistic dimension of anxiety (extracted from CSAI inventory) is again under debate [45]. Noteworthy, signaling from dopamine neurons of the ventral tegmental area (VTA) in the mesoaccumbal and mesocortical pathways of murine models has been shown to play a vital role in anxiety behavior [46]. The ventral teg-mental area (VTA) is best known for its robust dopaminergic projections to forebrain regions and their critical role in regulating reward, motivation, cognition, and aversion [47]. Therefore, several studies are now focusing on dopaminergic circuits (also in VTA segments) to address the self-confidence component of sports anxiety, in parallel to those cortisol-mediated processes [48]”.
Moreover, it is important to emphasise what your research adds to the scientific community or experts in combat sports. In other words, how are these results significant for psychologists or coaches?
ANSWER: Regarding reviewer #2 comments, we decided to include the following sentences in the Conclusion session: (lines 430-436) “Therefore, plasmatic uric acid could be suggested as a potential biomarker of the anticipatory stress (including relevant anxiety dimensions) experienced by combat athletes before competitions, independently to psychological questionnaires. Based on that, sports psychologists, coaches, and sport physicians would be able to adapt training/polishing protocols, mindfulness sessions and all other circumstances preceding competitions to minimize stress, particularly on their more susceptible athletes”.
It seems you reported the limitations in the text; however, I suggest creating a new section to discuss the possible limitations of the study.
ANSWER: We would like to thank reviewer #2 for his/her pertinent suggestions, but we opted to stick to the format of Brain Science journal, which does not include a particular section for scientific limitations of the study. Instead, based on the pertinent suggestions, we decided to add more experimental limitations of our study in the Conclusion section, which now stands as:
(lines 436-441) “It should be kept in mind that we could not evidence the significant correlation be-tween cortisol and uric acid (or their pre/post variations). Moreover, this intervention-al study was conducted with a limited number of participants and the cause-consequence association between uric acid and psychopathology of anxiety in combat sports cannot be directly inferred from the current study, but it definitely de-serves further investigation”.
Minor points
The procedures are integrated into Psychological Analysis and Blood sampling section. I wondered if it is better to create a new section where you can explain the procedures.
For instance, could the "Biochemical assays" section be set as a subchapter of the Blood sampling section?
ANSWER: We understand reviewer #2 concerns but we considered important to discriminate “blood sampling” to “biochemical assays” procedures, eventhough both procedures use Chemical compounds and different laboratory equipments. Nevertheless, all experimental protocols were meticulously described here. With all respect to reviewer #2 suggestions, we prefer to keep these sections as they were presented in the original version.
I would change Psychological Analysis to Psychological Tools (instruments).
ANSWER: Pursuant to suggestions by reviewer #2, we corrected the text. Thank you.

Reviewer 3 Report
The manuscript of of Luis Fernando Garcia de Oliveira et al. entitled Uric Acid and Cortisol Levels in Plasma Correlate with Pre-Competition Anxiety in Novice Athletes of Combat Sports is interesting, but some mistakes have to be corrected before acceptance. Additionally, I have some minor comments as well.
Major:
1. The presentation of statistical data should be checked again through the entire manuscript.
E.g. Pearson’s r and p values are written oppositely in the text than in the table and on the figure at least 3 times in 3.2.
„Cortisol levels were linearly correlated with cognitive anxiety, as estimated by CSAI tests: Pearson’s r = 0.002, p-value = 0.766, and a strong Bayesian inference; BF10 = 22.17.”
Based on Fig. 2 and Table 4, r should be 0.766 and p should be 0.002)
„Regarding the inventories, CSAI-somatic anxiety was both correlated with CSAI-cognitive anxiety (Pearson’s r = 0.002, p-value = 0.762…”
Same problem.
„and with BAI scores (Pearson’s r = 0.015, p-value = 0.655…”
Same problem.
2. Figure 2 C: the negative sign is missing (r should be -0.324 like in Table 4).
Minor:
3. It should be indicated in the title or at least in the abstract, that male athletes have been examined.
4. „Subjects were not taking any antioxidant or multivitamin supplements at the time of the study and their diets did not contain more than 4,000 cal/day or a protein intake >1.75 g/kg body mass (Exclusion criterion 1).” It should be indicated, since when (like 2 months in case of Exclusion criterion 2).
Author Response
Comments and Suggestions for Authors
The manuscript of of Luis Fernando Garcia de Oliveira et al. entitled Uric Acid and Cortisol Levels in Plasma Correlate with Pre-Competition Anxiety in Novice Athletes of Combat Sports is interesting, but some mistakes have to be corrected before acceptance. Additionally, I have some minor comments as well.
Major:
- The presentation of statistical data should be checked again through the entire manuscript. E.g. Pearson’s r and p values are written oppositely in the text than in the table and on the figure at least 3 times in 3.2.
Cortisol levels were linearly correlated with cognitive anxiety, as estimated by CSAI tests: Pearson’s r = 0.002, p-value = 0.766, and a strong Bayesian inference; BF10 = 22.17.”
Based on Fig. 2 and Table 4, r should be 0.766 and p should be 0.002)
„Regarding the inventories, CSAI-somatic anxiety was both correlated with CSAI-cognitive anxiety (Pearson’s r = 0.002, p-value = 0.762…”. Same problem.
„and with BAI scores (Pearson’s r = 0.015, p-value = 0.655…” Same problem.
ANSWER: We are thankful to reviewer #3 for his/her observations. This is the kind of mistake that seriously depreciate the scientific quality of papers. We revised the whole manuscript for proper results’ presentation in this updated version. Thank you.
- Figure 2 C: the negative sign is missing (r should be -0.324 like in Table 4).
ANSWER: Pursuant to suggestions by reviewer #3, we corrected the text. Thank you.
Minor:
- It should be indicated in the title or at least in the abstract, that male athletes have been examined.
ANSWER: Pursuant to suggestions by reviewer #2, we corrected the text. Thank you.
- „Subjects were not taking any antioxidant or multivitamin supplements at the time of the study and their diets did not contain more than 4,000 cal/day or a protein intake >1.75 g/kg body mass (Exclusion criterion 1).” It should be indicated, since when (like 2 months in case of Exclusion criterion 2).
ANSWER: Pursuant to suggestions by reviewer #2, we corrected the text. Thank you.
We trust the modifications herewith will respond the major concerns of the referees and become more fluent and scientifically accurate.
I look forward to hearing from you. Sincerely yours,
Marcelo Barros

Round 2
Reviewer 2 Report
Dear Authors
Thank you for your detailed replies.
I think now the manuscript is enough improved.
However, there are other minor points to review:
ll 44 - 47. You wrote: "Many researchers ....", but you cited two investigations. Please cite more studies or a review, systematic review, or meta-analysis. If possible. Moreover, it should be better to write "[e.g., 2,3, etc..]" or something like that.
ll 132 - 138. It is ok, but it seems a reply to the other review. Please, rewrite in order to fit with the text.
In general the new entries are ok, but they should be rewritten to fit better with the text.
I also suggest a proof reading by a english native speaker. Some sentences need to be linked better.
Author Response
Dear Authors
Thank you for your detailed replies.
I think now the manuscript is enough improved.
However, there are other minor points to review:
ll 44 - 47. You wrote: "Many researchers ....", but you cited two investigations. Please cite more studies or a review, systematic review, or meta-analysis. If possible. Moreover, it should be better to write "[e.g., 2,3, etc..]" or something like that.
ANSWER: We fully agree with referee #2. Therefore, we decided to replace the original references from previous versions of the MS by meta-analysis studies and one more robust/recent work here.
ll 132 - 138. It is ok, but it seems a reply to the other review. Please, rewrite in order to fit with the text.
ANSWER: As suggested by reviewer #2, we properly rewrote the sentences from lines 132-138, which now stand as: “Although we expected that the anxiety level would increase later on, logistic issues (meeting for coach instructions, warming up, etc.) limited our further access to the athletes. We are aware of adapted stress/questionnaires to evaluate immediately pre-competition stress (e.g. CSAI-30), but we were not allowed to approach the athletes under these circumstances [26]”.
In general the new entries are ok, but they should be rewritten to fit better with the text. I also suggest a proof reading by a english native speaker. Some sentences need to be linked better.
ANSWER: Pursuant to reviewer #2 suggestions, we double-checked this updated version o four MS and corrections are marked in BLUE within.
We trust the modifications herewith will respond the major concerns of the reviewers and become more fluent and scientifically accurate.
I look forward to hearing from you.
Sincerely yours,
Marcelo Barros
